# Binarized Neural Networks

**Itay Hubara**[1]*
itayh@technion.ac.il

**Matthieu Courbariaux**[2]*
matthieu.courbariaux@gmail.com

**Daniel Soudry**[3]
daniel.soudry@gmail.com

**Ran El-Yaniv**[1]
rani@cs.technion.ac.il

**Yoshua Bengio**[2,4]
yoshua.umontreal@gmail.com

(1) Technion, Israel Institute of Technology.
(3) Columbia University.
(*) Indicates equal contribution.

(2) Université de Montréal.
(4) CIFAR Senior Fellow.

## Abstract

We introduce a method to train Binarized Neural Networks (BNNs) - neural networks with binary weights and activations at run-time. At train-time the binary weights and activations are used for computing the parameter gradients. During the forward pass, BNNs drastically reduce memory size and accesses, and replace most arithmetic operations with bit-wise operations, which is expected to substantially improve power-efficiency. To validate the effectiveness of BNNs, we conducted two sets of experiments on the Torch7 and Theano frameworks. On both, BNNs achieved nearly state-of-the-art results over the MNIST, CIFAR-10 and SVHN datasets. We also report our preliminary results on the challenging ImageNet dataset. Last but not least, we wrote a binary matrix multiplication GPU kernel with which it is possible to run our MNIST BNN 7 times faster than with an unoptimized GPU kernel, without suffering any loss in classification accuracy. The code for training and running our BNNs is available on-line.

## Introduction

Deep Neural Networks (DNNs) have substantially pushed Artificial Intelligence (AI) limits in a wide range of tasks (LeCun et al., 2015). Today, DNNs are almost exclusively trained on one or many very fast and power-hungry Graphic Processing Units (GPUs) (Coates et al., 2013). As a result, it is often a challenge to run DNNs on target low-power devices, and substantial research efforts are invested in speeding up DNNs at run-time on both general-purpose (Gong et al., 2014; Han et al., 2015b) and specialized computer hardware (Chen et al., 2014; Esser et al., 2015).

This paper makes the following contributions:

- We introduce a method to train Binarized-Neural-Networks (BNNs), neural networks with binary weights and activations, at run-time, and when computing the parameter gradients at train-time (see Section 1).

- We conduct two sets of experiments, each implemented on a different framework, namely Torch7 and Theano, which show that it is possible to train BNNs on MNIST, CIFAR-10 and SVHN and achieve near state-of-the-art results (see Section 2). Moreover, we report preliminary results on the challenging ImageNet dataset

- We show that during the forward pass (both at run-time and train-time), BNNs drastically reduce memory consumption (size and number of accesses), and replace most arithmetic operations with bit-wise operations, which potentially lead to a substantial increase in power-efficiency (see Section

3). Moreover, a binarized CNN can lead to binary convolution kernel repetitions; we argue that dedicated hardware could reduce the time complexity by $60\%$ .

- Last but not least, we programed a binary matrix multiplication GPU kernel with which it is possible to run our MNIST BNN 7 times faster than with an unoptimized GPU kernel, without suffering any loss in classification accuracy (see Section 4).

The code for training and running our BNNs is available on-line (both Theano[1] and Torch framework[2]).

# 1   Binarized Neural Networks

In this section, we detail our binarization function, show how we use it to compute the parameter gradients,and how we backpropagate through it.

**Deterministic vs Stochastic Binarization**   When training a BNN, we constrain both the weights and the activations to either $+1$ or $-1$. Those two values are very advantageous from a hardware perspective, as we explain in Section 4. In order to transform the real-valued variables into those two values, we use two different binarization functions, as in (Courbariaux et al., 2015). Our first binarization function is deterministic:

$$x^b = \text{Sign}(x) = \begin{cases} +1 & \text{if } x \geq 0, \\ -1 & \text{otherwise,} \end{cases} \tag{1}$$

where $x^b$ is the binarized variable (weight or activation) and $x$ the real-valued variable. It is very straightforward to implement and works quite well in practice. Our second binarization function is stochastic:

$$x^b = \begin{cases} +1 & \text{with probability } p = \sigma(x), \\ -1 & \text{with probability } 1 - p, \end{cases} \tag{2}$$

where $\sigma$ is the *"hard sigmoid"* function:

$$\sigma(x) = \text{clip}(\frac{x+1}{2}, 0, 1) = \max(0, \min(1, \frac{x+1}{2})). \tag{3}$$

The stochastic binarization is more appealing than the sign function, but harder to implement as it requires the hardware to generate random bits when quantizing. As a result, we mostly use the deterministic binarization function (i.e., the sign function), with the exception of *activations at train-time* in some of our experiments.

**Gradient Computation and Accumulation**   Although our BNN training method uses binary weights and activation to compute the parameter gradients, the real-valued gradients of the weights are accumulated in real-valued variables, as per Algorithm 1. Real-valued weights are likely required for Stochasic Gradient Descent (SGD) to work at all. SGD explores the space of parameters in small and noisy steps, and that noise is *averaged out* by the stochastic gradient contributions accumulated in each weight. Therefore, it is important to maintain sufficient resolution for these accumulators, which at first glance suggests that high precision is absolutely required.

Moreover, adding noise to weights and activations when *computing* the parameter gradients provide a form of regularization that can help to generalize better, as previously shown with variational weight noise (Graves, 2011), Dropout (Srivastava et al., 2014) and DropConnect (Wan et al., 2013). Our method of training BNNs can be seen as a variant of Dropout, in which instead of randomly setting half of the activations to zero when computing the parameter gradients, we binarize both the activations and the weights.

**Propagating Gradients Through Discretization**   The derivative of the sign function is zero almost everywhere, making it apparently incompatible with back-propagation, since the exact gradient of the cost with respect to the quantities before the discretization (pre-activations or weights) would

be zero. Note that this remains true even if stochastic quantization is used. Bengio (2013) studied the question of estimating or propagating gradients through stochastic discrete neurons. He found in his experiments that the fastest training was obtained when using the "straight-through estimator," previously introduced in Hinton's lectures (Hinton, 2012). We follow a similar approach but use the version of the straight-through estimator that takes into account the saturation effect, and does use deterministic rather than stochastic sampling of the bit. Consider the sign function quantization

$$q = \text{Sign}(r),$$

and assume that an estimator $g_q$ of the gradient $\frac{\partial C}{\partial q}$ has been obtained (with the straight-through estimator when needed).

**Algorithm 1:** Training a BNN. $C$ is the cost function for minibatch, $\lambda$ the learning rate decay factor and $L$ the number of layers. $\circ$ indicates element-wise multiplication. The function Binarize() specifies how to (stochastically or deterministically) binarize the activations and weights, and Clip() specifies how to clip the weights. BatchNorm() specifies how to batch-normalize the activations, using either batch normalization (Ioffe & Szegedy, 2015) or its shift-based variant we describe in Algorithm 3. BackBatchNorm() specifies how to back-propagate through the normalization. Update() specifies how to update the parameters when their gradients are known, using either ADAM (Kingma & Ba, 2014) or the shift-based AdaMax we describe in Algorithm 2.

**Require:** a minibatch of inputs and targets $(a_0, a^*)$, previous weights $W$, previous BatchNorm parameters $\theta$, weight initialization coefficients from (Glorot & Bengio, 2010) $\gamma$, and previous learning rate $\eta$.
**Ensure:** updated weights $W^{t+1}$, updated BatchNorm parameters $\theta^{t+1}$ and updated learning rate $\eta^{t+1}$.
{1. Computing the gradients:}
{1.1. Forward propagation:}
**for** $k = 1$ to $L$ **do**
  $W_k^b \leftarrow \text{Binarize}(W_k)$, $s_k \leftarrow a_{k-1}^b W_k^b$
  $a_k \leftarrow \text{BatchNorm}(s_k, \theta_k)$
  **if** $k < L$ **then** $a_k^b \leftarrow \text{Binarize}(a_k)$
{1.2. Backward propagation:}
{Please note that the gradients are not binary.}
Compute $g_{a_L} = \frac{\partial C}{\partial a_L}$ knowing $a_L$ and $a^*$
**for** $k = L$ to $1$ **do**
  **if** $k < L$ **then** $g_{a_k} \leftarrow g_{a_k^b} \circ 1_{|a_k| \leq 1}$
  $(g_{s_k}, g_{\theta_k}) \leftarrow \text{BackBatchNorm}(g_{a_k}, s_k, \theta_k)$
  $g_{a_{k-1}^b} \leftarrow g_{s_k} W_k^b$, $g_{W_k^b} \leftarrow g_{s_k}^\top a_{k-1}^b$
{2. Accumulating the gradients:}
**for** $k = 1$ to $L$ **do**
  $\theta_k^{t+1} \leftarrow \text{Update}(\theta_k, \eta^t, g_{\theta_k})$, $\eta^{t+1} \leftarrow \lambda \eta^t$
  $W_k^{t+1} \leftarrow \text{Clip}(\text{Update}(W_k, \gamma_k \eta^t, g_{W_k^b}), -1, 1)$

**Algorithm 2:** Shift based AdaMax learning rule (Kingma & Ba, 2014). $g_t^2$ indicates the element-wise square $g_t \circ g_t$ and $\oslash$ stands for **both** left and right bit-shift. Good default settings are $\alpha = 2^{-10}$, $1 - \beta_1 = 2^{-3}$, $1 - \beta_2 = 2^{-10}$. All operations on vectors are element-wise. With $\beta_1^t$ and $\beta_2^t$ we denote $\beta_1$ and $\beta_2$ to the power $t$.

**Require:** Previous parameters $\theta_{t-1}$ and their gradient $g_t$, and learning rate $\alpha$.
**Ensure:** Updated parameters $\theta_t$.
{Biased 1st and 2nd moment estimates:}

$$m_t \leftarrow \beta_1 \cdot m_{t-1} + (1 - \beta_1) \cdot g_t$$
$$v_t \leftarrow \max(\beta_2 \cdot v_{t-1}, |g_t|)$$
{Updated parameters:}
$$\theta_t \leftarrow \theta_{t-1} - (\alpha \oslash (1 - \beta_1)) \cdot \hat{m} \oslash v_t^{-1}$$

**Algorithm 3:** Shift based Batch Normalizing Transform, applied to activation x over a mini-batch. The approximate power-of-2 is[3] $AP2(x) = \text{sign}(x) 2^{\text{round}(\log 2 |x|)}$, and $\oslash$ stands for **both** left and right binary shift.

**Require:** Values of $x$ over a mini-batch: $B = \{x_{1...m}\}$; parameters to learn: $\gamma, \beta$.
**Ensure:** $\{y_i = \text{BN}(x_i, \gamma, \beta)\}$
{1. Mini-batch mean:}
$\mu_B \leftarrow \frac{1}{m} \sum_{i=1}^m x_i$
{2. Centered input: }
$C(x_i) \leftarrow (x_i - \mu_B)$
{3. Approximate variance:}
$\sigma_B^2 \leftarrow \frac{1}{m} \sum_{i=1}^m (C(x_i) \oslash AP2(C(x_i)))$
{4. Normalize:}
$\hat{x_i} \leftarrow C(x_i) \oslash AP2((\sqrt{\sigma_B^2 + \epsilon})^{-1})$
{5. Scale and shift:}
$y_i \leftarrow AP2(\gamma) \oslash \hat{x_i}$

Then, our straight-through estimator of $\frac{\partial C}{\partial r}$ is simply

$$g_r = g_q 1_{|r| \leq 1}. \tag{4}$$

Note that this preserves the gradient's information and cancels the gradient when $r$ is too large. Not cancelling the gradient when $r$ is too large significantly worsens the performance. The use of this straight-through estimator is illustrated in Algorithm 1. The derivative $1_{|r| \leq 1}$ can also be seen as propagating the gradient through *hard tanh*, which is the following piece-wise linear activation function:

$$\text{Htanh}(x) = \text{Clip}(x, -1, 1). \tag{5}$$

For hidden units, we use the sign function non-linearity to obtain binary activations, and for weights we combine two ingredients:

- Constrain each real-valued weight between -1 and 1, by projecting $w^r$ to -1 or 1 when the weight update brings $w^r$ outside of $[-1, 1]$, i.e., clipping the weights during training, as per Algorithm 1. The real-valued weights would otherwise grow very large without any impact on the binary weights.

- When using a weight $w^r$, quantize it using $w^b = \text{Sign}(w^r)$.

This is consistent with the gradient canceling when $|w^r| > 1$, according to Eq. 4.

**Algorithm 4:** Running a BNN. $L$ = layers.

**Require:** a vector of 8-bit inputs $a_0$, the binary weights $W^b$, and the BatchNorm parameters $\theta$.
**Ensure:** the MLP output $a_L$.
{1. First layer:}
$a_1 \leftarrow 0$
**for** $n = 1$ to 8 **do**
$\quad a_1 \leftarrow a_1 + 2^{n-1} \cdot \text{XnorDotProduct}(a_0^n, W_1^b)$
$a_1^b \leftarrow \text{Sign}(\text{BatchNorm}(a_1, \theta_1))$
{2. Remaining hidden layers:}
**for** $k = 2$ to $L - 1$ **do**
$\quad a_k \leftarrow \text{XnorDotProduct}(a_{k-1}^b, W_k^b)$
$\quad a_k^b \leftarrow \text{Sign}(\text{BatchNorm}(a_k, \theta_k))$
{3. Output layer:}
$a_L \leftarrow \text{XnorDotProduct}(a_{L-1}^b, W_L^b)$
$a_L \leftarrow \text{BatchNorm}(a_L, \theta_L)$

**Shift-based Batch Normalization** Batch Normalization (BN) (Ioffe & Szegedy, 2015), accelerates the training and also seems to reduces the overall impact of the weight scale. The normalization noise may also help to regularize the model. However, at train-time, BN requires many multiplications (calculating the standard deviation and dividing by it), namely, dividing by the running variance (the weighted mean of the training set activation variance). Although the number of scaling calculations is the same as the number of neurons, in the case of ConvNets this number is quite large. For example, in the CIFAR-10 dataset (using our architecture), the first convolution layer, consisting of only $128 \times 3 \times 3$ filter masks, converts an image of size $3 \times 32 \times 32$ to size $3 \times 128 \times 28 \times 28$, which is two orders of magnitude larger than the number of weights. To achieve the results that BN would obtain, we use a shift-based batch normalization (SBN) technique. detailed in Algorithm 3. SBN approximates BN almost without multiplications. In the experiment we conducted we did not observe accuracy loss when using the shift based BN algorithm instead of the vanilla BN algorithm.

**Shift based AdaMax** The ADAM learning rule (Kingma & Ba, 2014) also seems to reduce the impact of the weight scale. Since ADAM requires many multiplications, we suggest using instead the shift-based AdaMax we detail in Algorithm 2. In the experiment we conducted we did not observe accuracy loss when using the shift-based AdaMax algorithm instead of the vanilla ADAM algorithm.

**First Layer** In a BNN, only the binarized values of the weights and activations are used in all calculations. As the output of one layer is the input of the next, all the layers inputs are binary, with the exception of the first layer. However, we do not believe this to be a major issue. First, in computer vision, the input representation typically has far fewer channels (e.g, red, green and blue) than internal representations (e.g, 512). As a result, the first layer of a ConvNet is often the smallest convolution layer, both in terms of parameters and computations (Szegedy et al., 2014). Second, it is relatively easy to handle continuous-valued inputs as fixed point numbers, with $m$ bits of precision. For example, in the common case of 8-bit fixed point inputs:

$$ s = x \cdot w^b \qquad ; \qquad s = \sum_{n=1}^{8} 2^{n-1}(x^n \cdot w^b), \qquad (6) $$

where $x$ is a vector of 1024 8-bit inputs, $x_1^8$ is the most significant bit of the first input, $w^b$ is a vector of 1024 1-bit weights, and $s$ is the resulting weighted sum. This trick is used in Algorithm 4.

## 2 Benchmark Results

We conduct two sets of experiments, each based on a different framework, namely Torch7 and Theano. Implementation details are reported in Appendix A and code for both frameworks is available online. Results are reported in Table 1.

**Table 1:** Classification test error rates of DNNs trained on MNIST (fully connected architecture), CIFAR-10 and SVHN (convnet). No unsupervised pre-training or data augmentation was used.

| Data set | MNIST | SVHN | CIFAR-10 |
|---|---|---|---|
| Binarized activations+weights, during training and test | | | |
| BNN (Torch7) | 1.40% | 2.53% | 10.15% |
| BNN (Theano) | 0.96% | 2.80% | 11.40% |
| Committee Machines' Array (Baldassi et al., 2015) | 1.35% | - | - |
| Binarized weights, during training and test | | | |
| BinaryConnect (Courbariaux et al., 2015) | $1.29\pm 0.08\%$ | 2.30% | 9.90% |
| Binarized activations+weights, during test | | | |
| EBP (Cheng et al., 2015) | $2.2\pm 0.1\%$ | - | - |
| Bitwise DNNs (Kim & Smaragdis, 2016) | 1.33% | - | - |
| Ternary weights, binary activations, during test | | | |
| (Hwang & Sung, 2014) | 1.45% | - | - |
| No binarization (standard results) | | | |
| No regularization | $1.3\pm 0.2\%$ | 2.44% | 10.94% |
| Gated pooling (Lee et al., 2015) | - | 1.69% | 7.62% |

**Preliminary Results on ImageNet** To test the strength of our method, we applied it to the challenging ImageNet classification task. Considerable research has been concerned with compressing ImageNet architectures while preserving high accuracy performance (e.g., Han et al. (2015a)). Previous approaches that have been tried include pruning near zero weights using matrix factorization techniques, quantizing the weights and applying Huffman codes among others. To the best of the our knowledge, so far there are no reports on successfully quantizing the network's activations. Moreover, a recent work Han et al. (2015a) showed that accuracy significantly deteriorates when trying to quantize convolutional layers' weights below 4 bits (FC layers are more robust to quantization and can operate quite well with only 2 bits). In the present

**Figure 1:** Training curves for different methods on CIFAR-10 dataset. The dotted lines represent the training costs (square hinge losses) and the continuous lines the corresponding validation error rates. Although BNNs are slower to train, they are nearly as accurate as 32-bit float DNNs.

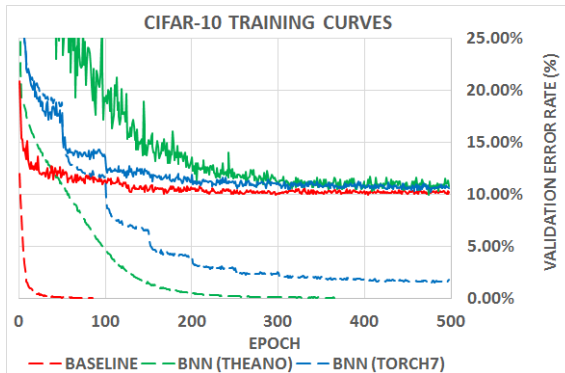

work we attempted to tackle the difficult task of binarizing both weights and activations. Employing the well known AlexNet and GoogleNet architectures, we applied our techniques and achieved 36.1% top-1 and 60.1% top-5 accuracies using AlexNet and 47.1% top-1 and 69.1% top-5 accuracies using GoogleNet. While this performance leaves room for improvement (relative to full precision nets), they are by far better than all previous attempts to compress ImageNet architectures using less than 4 bits precision for the weights. Moreover, this advantage is achieved while also binarizing neuron activations. Detailed descriptions of these results as well as full implementation details of our experiments are reported in the supplementary material (Appendix B). In our latest work (Hubara et al., 2016) we relaxed the binary constrains and allowed more than 1-bit per weight and activations. The resulting QNNs achieve prediction accuracy comparable to their 32-bit counterparts. For example, our quantized version of AlexNet with 1-bit weights and 2-bit activations achieves 51% top-1 accuracy and GoogleNet with 4-bits weighs and activation achived 66.6%. Moreover, we quantize the parameter gradients to 6-bits as well which enables gradients computation using only bit-wise operation. Full details can be found in (Hubara et al., 2016)

**Table 2:** Energy consumption of multiply-accumulations in pico-joules (Horowitz, 2014)

| Operation | MUL | ADD |
|---|---|---|
| 8bit Integer | 0.2pJ | 0.03pJ |
| 32bit Integer | 3.1pJ | 0.1pJ |
| 16bit Floating Point | 1.1pJ | 0.4pJ |
| 32tbit Floating Point | 3.7pJ | 0.9pJ |

**Table 3:** Energy consumption of memory accesses in pico-joules (Horowitz, 2014)

| Memory size | 64-bit memory access |
|---|---|
| 8K | 10pJ |
| 32K | 20pJ |
| 1M | 100pJ |
| DRAM | 1.3-2.6nJ |

## 3   High Power Efficiency during the Forward Pass

Computer hardware, be it general-purpose or specialized, is composed of memories, arithmetic operators and control logic. During the forward pass (both at run-time and train-time), BNNs drastically reduce memory size and accesses, and replace most arithmetic operations with bit-wise operations, which might lead to a great increase in power-efficiency. Moreover, a binarized CNN can lead to binary convolution kernel repetitions, and we argue that dedicated hardware could reduce the time complexity by $60\%$ .

**Memory Size and Accesses**   Improving computing performance has always been and remains a challenge. Over the last decade, power has been the main constraint on performance (Horowitz, 2014). This is why much research effort has been devoted to reducing the energy consumption of neural networks. Horowitz (2014) provides rough numbers for the energy consumed by the computation (the given numbers are for 45nm technology), as summarized in Tables 2 and 3. Importantly, we can see that memory accesses typically consume more energy than arithmetic operations, and *memory access cost augments with memory size*. In comparison with 32-bit DNNs, BNNs require 32 times smaller memory size *and* 32 times fewer memory accesses. This is expected to reduce energy consumption drastically (i.e., more than 32 times).

**XNOR-Count**   Applying a DNN mainly consists of convolutions and matrix multiplications. The key arithmetic operation of deep learning is thus the multiply-accumulate operation. Artificial neurons are basically multiply-accumulators computing weighted sums of their inputs. In BNNs, both the activations and the weights are constrained to either $-1$ or $+1$. As a result, most of the 32-bit floating point multiply-accumulations are replaced by 1-bit XNOR-count operations. This could have a big impact on dedicated deep learning hardware. For instance, a 32-bit floating point multiplier costs about 200 Xilinx FPGA slices (Govindu et al., 2004; Beauchamp et al., 2006), whereas a 1-bit XNOR gate only costs a single slice.

**Exploiting Filter Repetitions**   When using a ConvNet architecture with binary weights, the number of unique filters is bounded by the filter size. For example, in our implementation we use filters of size $3 \times 3$, so the maximum number of unique 2D filters is $2^9 = 512$. Since we now have binary filters, many 2D filters of size $k \times k$ repeat themselves. By using dedicated hardware/software, we can apply only the unique 2D filters on each feature map and sum the results to receive each 3D filter's convolutional result. For example, in our ConvNet architecture trained on the CIFAR-10 benchmark, there are only 42% unique filters per layer on average. Hence we can reduce the number of the XNOR-popcount operations by 3.

## 4   Seven Times Faster on GPU at Run-Time

It is possible to speed up GPU implementations of BNNs, by using a method sometimes called SIMD (single instruction, multiple data) within a register (SWAR). The basic idea of SWAR is to *concatenate* groups of 32 binary variables into 32-bit registers, and thus obtain a 32-times speed-up on bitwise operations (e.g, XNOR). Using SWAR, it is possible to evaluate 32 connections with only 3 instructions:

$$a_1+ = \mathrm{popcount}(\mathrm{xnor}(a_0^{32b}, w_1^{32b})), \tag{7}$$

where $a_1$ is the resulting weighted sum, and $a_0^{32b}$ and $w_1^{32b}$ are the concatenated inputs and weights. Those 3 instructions (accumulation, popcount, xnor) take $1 + 4 + 1 = 6$ clock cycles on recent

Nvidia GPUs (and if they were to become a fused instruction, it would only take a single clock cycle). Consequently, we obtain a theoretical Nvidia GPU speed-up of factor of $32/6 \approx 5.3$. In practice, this speed-up is quite easy to obtain as the memory bandwidth to computation ratio is also increased by 6 times.

In order to validate those theoretical results, we programed two GPU kernels:

- The first kernel (baseline) is an unoptimized matrix multiplication kernel.

- The second kernel (XNOR) is nearly identical to the baseline kernel, except that it uses the SWAR method, as in Equation (7).

The two GPU kernels return identical outputs when their inputs are constrained to $-1$ or $+1$ (but not otherwise). The XNOR kernel is about *23 times faster than the baseline kernel* and *3.4 times faster than cuBLAS*, as shown in Figure 2. Last but not least, the MLP from Section 2 runs 7 times faster with the XNOR kernel than with the baseline kernel, without suffering any loss in classification accuracy (see Figure 2).

## 5 Discussion and Related Work

Until recently, the use of extremely low-precision networks (binary in the extreme case) was believed to be highly destructive to the network performance (Courbariaux et al., 2014). Soudry et al. (2014) and Cheng et al. (2015) proved the contrary by showing that good performance could be achieved even if all neurons and weights are binarized to $\pm 1$ . This was done using Expectation BackPropagation (EBP), a variational Bayesian approach, which infers net-

**Figure 2:** The first three columns represent the time it takes to perform a $8192 \times 8192 \times 8192$ (binary) matrix multiplication on a GTX750 Nvidia GPU, depending on which kernel is used. We can see that our XNOR kernel is 23 times faster than our baseline kernel and 3.4 times faster than cuBLAS. The next three columns represent the time it takes to run the MLP from Section 2 on the full MNIST test set. As MNIST's images are not binary, the first layer's computations are always performed by the baseline kernel. The last three columns show that the MLP accuracy does not depend on which kernel is used.

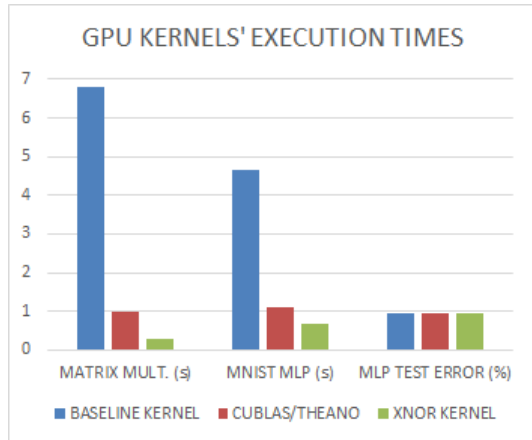

works with binary weights and neurons by updating the posterior distributions over the weights. These distributions are updated by differentiating their parameters (e.g., mean values) via the back propagation (BP) algorithm. Esser et al. (2015) implemented a fully binary network at run time using a very similar approach to EBP, showing significant improvement in energy efficiency. The drawback of EBP is that the binarized parameters are only used during inference.

The probabilistic idea behind EBP was extended in the BinaryConnect algorithm of Courbariaux et al. (2015). In BinaryConnect, the real-valued version of the weights is saved and used as a key reference for the binarization process. The binarization noise is independent between different weights, either by construction (by using stochastic quantization) or by assumption (a common simplification; see Spang (1962). The noise would have little effect on the next neuron's input because the input is a summation over many weighted neurons. Thus, the real-valued version could be updated by the back propagated error by simply ignoring the binarization noise in the update. Using this method, Courbariaux et al. (2015) were the first to binarize weights in CNNs and achieved near state-of-the-art performance on several datasets. They also argued that noisy weights provide a form of regularization, which could help to improve generalization, as previously shown in (Wan et al., 2013). This method binarized weights while still maintaining full precision neurons.

Lin et al. (2015) carried over the work of Courbariaux et al. (2015) to the back-propagation process by quantizing the representations at each layer of the network, to convert some of the remaining multiplications into bit-shifts by restricting the neurons values to be power-of-two integers. Lin et al. (2015)'s work and ours seem to share similar characteristics . However, their approach continues to use full precision weights during the test phase. Moreover, Lin et al. (2015) quantize the neurons only during the back propagation process, and not during forward propagation.

Other research Baldassi et al. (2015) showed that full binary training and testing is possible in an array of committee machines with randomized input, where only one weight layer is being adjusted. Gong et al. (2014) aimed to compress a fully trained high precision network by using a quantization or matrix factorization methods. These methods required training the network with full precision weights and neurons, thus requiring numerous MAC operations the proposed BNN algorithm avoids. Hwang & Sung (2014) focused on a fixed-point neural network design and achieved performance almost identical to that of the floating-point architecture. Kim & Smaragdis (2016) *retrained* neural networks with binary weights and activations.

So far, to the best of our knowledge, no work has succeeded in binarizing weights *and* neurons, at the inference phase *and* the entire training phase of a deep network. This was achieved in the present work. We relied on the idea that binarization can be done stochastically, or be approximated as random noise. This was previously done for the weights by Courbariaux et al. (2015), but our BNNs extend this to the activations. Note that the binary activations are especially important for ConvNets, where there are typically many more neurons than free weights. This allows highly efficient operation of the binarized DNN at run time, and at the forward-propagation phase during training. Moreover, our training method has almost no multiplications, and therefore might be implemented efficiently in dedicated hardware. However, we have to save the value of the full precision weights. This is a remaining computational bottleneck during training, since it is an energy-consuming operation.

## Conclusion

We have introduced BNNs, which binarize deep neural networks and can lead to dramatic improvements in both power consumption and computation speed. During the forward pass (both at run-time and train-time), BNNs drastically reduce memory size and accesses, and replace most arithmetic operations with bit-wise operations. Our estimates indicate that power efficiency can be improved by more than one order of magnitude (see Section 3). In terms of speed, we programed a binary matrix multiplication GPU kernel that enabled running MLP over the MNIST datset 7 times faster (than with an unoptimized GPU kernel) without suffering any accuracy degradation (see Section 4).

We have shown that BNNs can handle MNIST, CIFAR-10 and SVHN while achieving nearly state-of-the-art accuracy performance. While our preliminary results for the challenging ImageNet are not on par with the best results achievable with full precision networks, they significantly improve all previous attempts to compress ImageNet-capable architectures (see Section 2 and supplementary material - Appendix B). Moreover by relaxing the binary constrains and allowed more than 1-bit per weight and activations we have been able to achieve prediction accuracy comparable to their 32-bit counterparts. Full details can be found in our latest work (Hubara et al., 2016) A major open question would be to further improve our results on ImageNet. A substantial progress in this direction might lead to huge impact on DNN usability in low power instruments such as mobile phones.

## Acknowledgments

We would like to express our appreciation to Elad Hoffer, for his technical assistance and constructive comments. We thank our fellow MILA lab members who took the time to read the article and give us some feedback. We thank the developers of Torch, Collobert et al. (2011) a Lua based environment, and Theano (Bergstra et al., 2010; Bastien et al., 2012), a Python library which allowed us to easily develop a fast and optimized code for GPU. We also thank the developers of Pylearn2 (Goodfellow et al., 2013) and Lasagne (Dieleman et al., 2015), two Deep Learning libraries built on the top of Theano. We thank Yuxin Wu for helping us compare our GPU kernels with cuBLAS. We are also grateful for funding from NSERC, the Canada Research Chairs, Compute Canada, and CIFAR. We are also grateful for funding from CIFAR, NSERC, IBM, Samsung. This research was also supported by The Israel Science Foundation (grant No. 1890/14).

## Footnotes

[1]https://github.com/MatthieuCourbariaux/BinaryNet

[2]https://github.com/itayhubara/BinaryNet

[3]Hardware implementation of AP2 is as simple as extracting the index of the most significant bit from the number's binary representation.

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
