[Supplementary Material]

# Supplementary Material for "Binarized Neural Networks"

**Itay Hubara**[1]*
itayh@technion.ac.il

**Matthieu Courbariaux**[2]*
matthieu.courbariaux@gmail.com

**Daniel Soudry**[3]
daniel.soudry@gmail.com

**Ran El-Yaniv**[1]
rani@cs.technion.ac.il

**Yoshua Bengio**[2,4]
yoshua.umontreal@gmail.com

(1) Technion, Israel Institute of Technology.   (2) Université de Montréal.
(3) Columbia University.   (4) CIFAR Senior Fellow.
(*) Indicates equal contribution.

*Indicates equal contribution.

## A    Benchmark Implementation Details

Two sets of experiments were conducted, each based on a different framework, namely Torch7 (Collobert et al., 2011) and Theano (Bergstra et al., 2010; Bastien et al., 2012). With the exception of the following main differences (and some other specific differences listed below), the two sets are quite similar:

- In our Torch7 experiments, the activations are *stochastically* binarized at train-time, whereas in our Theano experiments they are *deterministically* binarized.

- In our Torch7 experiments, we use the *shift-based AdaMax and BN* variants, which are detailed in Algorithms 2 and 3, whereas in our Theano experiments, we use *vanilla BN and ADAM* .

In both sets of experiments, we obtained near state-of-the-art results with BNNs over the MNIST, CIFAR-10 and SVHN benchmark datasets.

### A.1    MLP on MNIST (Theano)

MNIST is an image classification benchmark dataset (LeCun et al., 1998). It consists of a training set containing 60K labeled instances, and a test set containing 10K instances. Each instance is a $28 \times 28$ gray-scale image of a hand-written digit in $\{0, 1 \ldots, 9\}$. In order for this benchmark to remain a challenge, we did not employ any convolution, data-augmentation, unsupervised feature generation or any other preprocessing. The MLP we trained on MNIST consists of 3 hidden layers each containing 4096 binary units and a L2-SVM output layer; we note that L2-SVM has been shown to perform better than Softmax on several classification benchmarks (Tang, 2013; Lee et al., 2014). We regularize the model with Dropout (Srivastava, 2013; Srivastava et al., 2014). The square hinge loss is minimized with the ADAM adaptive learning rate method (Kingma & Ba, 2014b). We use an exponentially decaying global learning rate, as per Algorithm 1, and also scale the learning rates of the weights with their initialization coefficients from (Glorot & Bengio, 2010), as suggested by Courbariaux et al. (2015a). We use Batch Normalization with a minibatch of size 100 to speed up the training. We follow the convention of utilizing the last 10K training set samples as a validation set for early stopping and model selection. We report the test error rate associated with the best validation error rate after 1000 epochs (we do not retrain on the validation set).

Table 1: Architecture of our CIFAR-10 ConvNet. We only use "same" convolutions, as in VGG (Simonyan & Zisserman, 2015).

| CIFAR-10 ConvNet architecture |
| --- |
| Input: $32 \times 32$ - RGB image |
| $3 \times 3$ - 128 convolution layer |
| BatchNorm and Binarization layers |
| $3 \times 3$ - 128 convolution and $2 \times 2$ max-pooling layers |
| BatchNorm and Binarization layers |
| $3 \times 3$ - 256 convolution layer |
| BatchNorm and Binarization layers |
| $3 \times 3$ - 256 convolution and $2 \times 2$ max-pooling layers |
| BatchNorm and Binarization layers |
| $3 \times 3$ - 512 convolution layer |
| BatchNorm and Binarization layers |
| $3 \times 3$ - 512 convolution and $2 \times 2$ max-pooling layers |
| BatchNorm and Binarization layers |
| 1024 fully connected layer |
| BatchNorm and Binarization layers |
| 1024 fully connected layer |
| BatchNorm and Binarization layers |
| 10 fully connected layer |
| BatchNorm layer (no binarization) |
| Cost: Mean square hinge loss |

## A.2 MLP on MNIST (Torch7)

We use a similar architecture as in our Theano experiments, without Dropout, and with 2048 binary units per layer instead of 4096. Additionally, we use the shift base AdaMax and BN (with a minibatch of size 100) instead of the vanilla implementations, to reduce the number of multiplications. Likewise, we decay the learning rate by using a 1-bit right shift every 10 epochs.

## A.3 ConvNet on CIFAR-10 (Theano)

CIFAR-10 is an image classification benchmark dataset. It consists of a training set of size 50K and a test set of size 10K, where instance are $32 \times 32$ color images representing airplanes, automobiles, birds, cats, deer, dogs, frogs, horses, ships and trucks. We do not use data-augmentation, which is known to be a game changer for this dataset (Graham, 2014). The architecture of our ConvNet is the same architecture used by Courbariaux et al. (2015b), except of course for the additional binarization of the activations. The architecture used by Courbariaux et al. (2015a) is itself mainly inspired by VGG (Simonyan & Zisserman, 2015). The square hinge loss is minimized with ADAM. We use an exponentially decaying learning rate, as we do for MNIST. We scale the learning rates of the weights with their initialization coefficients from (Glorot & Bengio, 2010). We use Batch Normalization with a minibatch of size 50 to speed up the training. We use the last 5000 samples of the training set as a validation set. We report the test error rate associated with the best validation error rate after 500 training epochs (we do not retrain on the validation set).

## A.4 ConvNet on CIFAR-10 (Torch7)

We use the same architecture as in our Theano experiments. We apply shift-based AdaMax and BN (with a minibatch of size 200) instead of the vanilla implementations to reduce the number of multiplications. Likewise, we decay the learning rate by using a 1-bit right shift every 50 epochs.

## A.5 ConvNet on SVHN

SVHN is also an image classification benchmark dataset. It consists of a training set of size 604K examples and a test set of size 26K, where instances are $32 \times 32$ color images representing digits ranging from 0 to 9. In both sets of experiments, we follow the same procedure used for the CIFAR-10 experiments, with a few notable exceptions: we use half the number of units in the convolution layers, and we train for 200 epochs instead of 500 (because SVHN is significantly larger than CIFAR-10).

## B   ImageNet Experiment

### B.1   The ImageNet Dataset

ImageNet is typically used as a classification task, and presently considered to be among the most important benchmarks. It consists of a training set of size 1.2M samples and test set of size 50K. Each instance is categorized into one of 1000 categories, which span objects, animals, scenes, and even some abstract shapes. It is customary to report classification results for ImageNet using two error measures: top-1 and top-5, where the top-5 error rate is the fraction of test images for which the correct label is not among the five labels considered most probable by the model.

### B.2   ImageNet results

**Related work:**   Recently considerable efforts have been invested in compressing neural network parameters. So far most network compression attempts considered an offline approach whereby a full precision network trained over ImageNet was compressed using various techniques. Among the prominent approaches that have been used are weight sharing (Chen et al., 2015), weight clipping and quantization (Gong et al., 2014). Our ImageNet result is the most extreme quantization attempt for this dataset where all weights are binarized. The most aggressive previous quantization of ImageNet considered the AlexNet architecture with 4-bit weight representation for the convolutional layers and 2-bit representation for the fully connected layers (Han & Dally, 2015). Their attempts to further compress the network resulted in drastic accuracy degradation. For example, they report that accuracy almost completely vanishes using full binary (1-bit) quantization.

A similar approach to taking advantage of binary representations of activations to speed-up computation was very recently presented by Rastegari et al. (2016), shortly after our own preliminary technical report on the same idea was posted on Arxiv. The Xnor-Net idea reported in (Rastegari et al., 2016) relies on a small modification to our algorithm where binary weights and inputs are multiplied by their $L_1$ norm. Implementation of this normalization requires many additional multiplications by a different scaling factor for each patch in each sample; see (Rastegari et al., 2016), Section 3.2 Eq. 10 and Figure 2. These extra multiplications prevent efficient implementations of XnorNets on known hardware designs.

**Results:**   We report results obtained using the well-known AlexNet and GoogleNet architectures. Results are reported in Table 2 and 3. In comparison to previous methods concerning aggressive compression to less than 4-bit weight representations, we achieve state-of-the-art results. Note, however, that in all our experiments with these architectures the training errors did not approach zero. This might indicate that different architectures would be beneficial.

### B.3   Implementation Details

**AlexNet:**   Our AlexNet implementation consists of 5 convolution layers followed by 3 fully connected layers (see Table 4). We optimized the negative log-liklihood loss function, used ADAM (Kingma & Ba, 2014a) for optimization, applied batch-normalization layers (with a minibatch of size 512). Likewise, we decayed the learning rate by 0.1 every 20 epochs.

**GoogleNet:**   Our GoogleNet implementation consists of 2 convolution layers followed by 10 inception layers, spacial-average-pooling and fully connected classifier. We additionally used 2 auxiliary classifiers. Here again, we optimized using ADAM the negative log-likelihood, used batch-normalization layers (with a minibatch of size 64), and decayed the learning rate by 0.1 every 10 epochs.

Table 2: Classification test error rates of AlexNet model trained on ImageNet 1000 classification task. No unsupervised pre-training or data augmentation was used.

| Data set | Top-1 | Top-5 |
|---|---|---|
| Binarized activations+weights, during training and test | | |
| BNN | 40.1% | 66.3% |
| Xnor-Nets (Rastegari et al., 2016) | 44.2% | 69.2% |
| Quantize weights, during test | | |
| Deep Compression 4/2bit (conv/FC layer) (Han et al., 2015) | 55.34% | 77.67% |
| (Gysel et al., 2016) - 2bit | 0.01% | -% |
| No Quantization (standard results) | | |
| AlexNet - our implementation | 56.6% | 80.2% |

Table 3: Classification test error rates of GoogleNet model trained on ImageNet 1000 classification task. No unsupervised pre-training or data augmentation was used.

| Data set | Top-1 | Top-5 |
|---|---|---|
| Binarized activations+weights, during training and test | | |
| BNN | 47.1% | 68.3% |
| No Quantization (standard results) | | |
| GoogleNet - our implementation | 71% | 90.2% |

Table 4: Architecture of our AlexNet.

| AlexNet ConvNet architecture |
|---|
| Input: $32 \times 32$ - RGB image |
| $11 \times 11$ - 64 convolution layer and $3 \times 3$ max-pooling layers |
| BatchNorm and Binarization layers |
| $5 \times 5$ - 192 convolution layer and $3 \times 3$ max-pooling layers |
| BatchNorm and Binarization layers |
| $3 \times 3$ - 384 convolution layer |
| BatchNorm and Binarization layers |
| $3 \times 3$ - 256 convolution layer |
| BatchNorm and Binarization layers |
| $3 \times 3$ - 256 convolution layer |
| BatchNorm and Binarization layers |
| 4096 fully connected layer |
| BatchNorm and Binarization layers |
| 4096 fully connected layer |
| BatchNorm and Binarization layers |
| 1000 fully connected layer |
| BatchNorm layer (no binarization) |
| SoftMax layer (no binarization) |
| Cost: Negative log likelihood |

## C  Exploiting Filter Repetitions

When using a ConvNet architecture with binary weights, the number of unique filters is bounded by the filter size. For example, in our implementation we use filters of size $3 \times 3$, so the maximum number of unique 2D filters is $2^9 = 512$. However, this should not prevent expanding the number of feature maps beyond this number, since the actual filter is a 3D matrix. Assuming we have $M_\ell$ filters in the $\ell$ convolutional layer, we have to store a 4D weight matrix of size $M_\ell \times M_{\ell-1} \times k \times k$. Consequently, the number of unique filters is $2^{k^2 M_{\ell-1}}$. When necessary, we apply each filter on the map and perform the required multiply-accumulate (MAC) operations (in our case, using XNOR and popcount operations). Since we now have binary filters, many 2D filters of size $k \times k$ repeat themselves. By using dedicated hardware/software, we can apply only the unique 2D filters on each feature map and sum the results to receive each 3D filter's convolutional result. Note that an inverse filter (i.e., [-1,1,-1] is the inverse of [1,-1,1]) can also be treated as a repetition; it is merely a multiplication of the original filter by -1. For example, in our ConvNet architecture trained on the CIFAR-10 benchmark, there are only 42% unique filters per layer on average. Hence we can reduce the number of the XNOR-popcount operations by 3.