[Reviews · NeurIPS 2016]

Reviewer 1

Summary

This paper proposed a method to train neural networks with binary weights and activations in order to reduce memory size and accelerate test speed.

Qualitative Assessment

The authors provide detailed theory to optimize binarized neural networks(BNNs) and extensive experiments to justify his theory. Binarization is an extreme case of the widely used quantization techique, but it still has some limitations. First, from information theory perspective, binarized neural networks have limited "knowledge" capacity which is not enough to deal with large-scale challenge. Also, optimization with sign function itself is an open challenging problem, though this paper proposed a reasonable approximation method. For small datasets it's possible to find a good local optima. But for large-scale datasets, it's quite easy to fall into a bad local optima using SGD and that might be why the results on ImageNet dataset are not promising. So to make a trade-off between speed and accuracy, why not add more bits on both weights and activations (or just on weights/activations)? OVERALL, I like the idea of binarizing deep neural networks but its ability to generalize to large-scale datasets is unconvincing.

Confidence in this Review

2-Confident (read it all; understood it all reasonably well)


Reviewer 2

Summary

The paper presents a method for training neural networks with binary weights and activations. It is essentially based on back-propagation where weights and activations are binarized during the forward pass, but the backward pass uses real-valued gradients and weight updates. The networks use batch normalisation and are optimised with Adam or AdaMax, which are also speeded-up by using binary shifts instead of multiplications. The framework is evaluated on MNIST, SVHN, CIFAR-10, and ImageNet where it performs worse than real-valued nets, but still demonstrates reasonable results. The authors also present the benchmarks of the GPU (CUDA) implementation of the binary matrix multiplication, which achieves a substantial speed-up over real-valued matrix multiplication.

Qualitative Assessment

Positives: 1) The topic of speeding-up training and evaluation of deep neural nets is very relevant, and the approach of moving the computationally intensive operations from real values to binary looks promising. 2) The method is evaluated on several datasets (including large-scale ImageNet). Negatives: 1) The contribution is rather incremental, the main difference with BinaryConnect is that the activations are also binarized, which requires a rather minor change in the algorithm. 2) Experimental evaluation is carried out using two different frameworks (Torch and Theano), with different networks, optimisers, binarization schemes, etc. As a result, the experimental section does not look cohesive and is hard to analyse or refer to in future work. Instead of providing the recipes for training BNNs, it confuses the reader. 3) The real-valued baselines for the same networks and optimisers are missing (apart from the ImageNet experiments). This is probably the most important question the experimental evaluation should answer -- how big is the performance gap between real-valued and binary nets, all other things being equal. Such baselines are only presented for ImageNet, where the gap is rather large, e.g. 71%->47.1% for GoogLeNet. It would be useful to see similar comparisons for other datasets. Questions: 1) For back-propagation through binarized activations, it is proposed to clip the gradients (eq. 5), but this approach is not used for back-propagation w.r.t. the binarized weights. Instead, it is effectively assumed that the gradient w.r.t. real-valued weights is the same as the gradient w.r.t. the binary weights g_{W_k^b}, but then there is clipping after the weight update. Why is back-propagation through binarized weights and activations done differently? Summary: Binary networks are a promising direction, and in principle the paper is interesting enough to be presented at NIPS, but the experimental evaluation does require improvement/re-writing. I would recommend the paper for borderline acceptance (conditioned on the authors improving the experimental section).

Confidence in this Review

3-Expert (read the paper in detail, know the area, quite certain of my opinion)


Reviewer 3

Summary

This paper presents neural networks with both binarized weights and activations. It includes the algorithms to train binarized neural networks with both deterministically binarized activations and stochastically binarized activations. Binarized neural networks of MNIST is 7 times faster than unoptimized GPU version without much loss in classification accuracy.

Qualitative Assessment

The results of BNN are promising that it can accelerate the speed of neural networks execution latency and reduce the memory and energy consumption. However, I have a few questions to this paper. First, it seems unclear to me why to use stochastic binarized activation. The results don't show consistent increase in accuracy by using stochastic activation while it requires the hardware support for generating random numbers. Second, compared to BinaryConnect which uses binarized weights, the benefit of binarized activation is not evaluated. It would be nice to see how much speed-up or energy reduction is brought by binarized activation. Also when compared to Xnor-net in ImageNet, BNN achieves significant lower accuracy than Xnor-net while the benefit of fewer multiplications in the normalization is not very clear. Third, table 2 and 3 is confusing. What is the number indicating in column 2 and 3 of table 2? Authors should provide descriptions for these two tables.

Confidence in this Review

2-Confident (read it all; understood it all reasonably well)